# Fluctuations in AKT and PTEN Activity Are Linked by the E3 Ubiquitin Ligase cCBL

**DOI:** 10.3390/cells10112803

**Published:** 2021-10-20

**Authors:** Manuel Olazábal-Morán, Miriam Sánchez-Ortega, Laura Martínez-Muñoz, Carmen Hernández, Manuel S. Rodríguez, Mario Mellado, Ana C. Carrera

**Affiliations:** 1Department of Immunology and Oncology, Centro Nacional de Biotecnología, CSIC-Campus Cantoblanco Autónoma University, 28049 Madrid, Spain; Olaza2004@gmail.com (M.O.-M.); msanchez@cnb.csic.es (M.S.-O.); laura.martinez@cabimer.es (L.M.-M.); mchdez@cnb.csic.es (C.H.); mmellado@cnb.csic.es (M.M.); 2Institut des Technologies Avancées en Sciences du Vivant (ITAV), 1 Place Pierre Potier, Université de Toulouse, CNRS, UPS, 31000 Toulouse, France; manuel.rodriguez@lcc-toulouse.fr

**Keywords:** PTEN, CBL, E3 ubiquitin ligase, phosphatidylinositide 3-kinase, AKT

## Abstract

3-Poly-phosphoinositides (PIP_3_) regulate cell survival, division, and migration. Both PI3-kinase (phosphoinositide-3-kinase) and PTEN (phosphatase and tensin-homolog in chromosome 10) control PIP_3_ levels, but the mechanisms connecting PI3-kinase and PTEN are unknown. Using non-transformed cells, the activation kinetics of PTEN and of the PIP_3_-effector AKT were examined after the addition of growth factors. Both epidermal growth factor and serum induced the early activation of AKT and the simultaneous inactivation of PTEN (at ~5 min). This PIP_3_/AKT peak was followed by a general reduction in AKT activity coincident with the recovery of PTEN phosphatase activity (at ~10–15 min). Subsequent AKT peaks and troughs followed. The fluctuation in AKT activity was linked to that of PTEN; PTEN reconstitution in PTEN-null cells restored AKT fluctuations, while PTEN depletion in control cells abrogated them. The analysis of PTEN activity fluctuations after the addition of growth factors showed its inactivation at ~5 min to be simultaneous with its transient ubiquitination, which was regulated by the ubiquitin E3 ligase cCBL (casitas B-lineage lymphoma proto-oncogene). Protein-protein interaction analysis revealed cCBL to be brought into the proximity of PTEN in a PI3-kinase-dependent manner. These results reveal a mechanism for PI3-kinase/PTEN crosstalk and suggest that cCBL could be new target in strategies designed to modulate PTEN activity in cancer.

## 1. Introduction

PIP_3_ (phosphoinositide 3,4,5P_3_), the product of PI3-kinase and its effector AKT, are essential mediators of cell survival [1,2,3,4]. While PIP_3_ formation is induced by PI3-kinase, PIP_3_ downregulation is caused by ‘phosphatase and tensin-homolog in chromosome 10’ (PTEN) [2,3,4]. Somatic mutations in *PTEN* are frequent in cancer, as is the loss of function of one *PTEN* allele. The latter is also seen in patients with hereditary PHTS syndrome (PTEN hamartoma tumor syndrome) who are more susceptible to the appearance of tumors [5,6,7].

The importance of PTEN function in cancer has prompted the study of the mechanisms altering its levels and activity, with most work performed in cancer cells. Certainly, PTEN function can be compromised by heterozygous gene loss, gene mutation, and by epigenetic, transcriptional, post-transcriptional, and post-translational modifications [5,6,7,8,9,10,11,12]. Post-translational modifications are particularly effective in inducing rapid and transient alterations in the architecture, localization, and activity of PTEN. These include C-terminal PTEN phosphorylation, which triggers an enzyme inactive conformation; PTEN sumoylation, which increases its localization to the plasma membrane (PM); and PTEN ubiquitination, which can influence PTEN stability, activity, and localization [10,11,12].

PTEN regulation has been studied extensively in cancer cells but little is known about how it is regulated in normal cells. The present work focuses on the analysis of wild type (WT) endogenous (endo)-PTEN regulation at near-physiological conditions by examining the changes in PTEN activity in response to cell activation by epidermal growth factor (EGF) or fetal calf serum [(FCS). Since PTEN action should, in theory, be connected to PI3-kinase (to control PIP_3_ levels), the activation kinetics of the PI3-kinase effector AKT was examined in parallel. The overall aim was to define how normal PTEN activity is regulated and to propose new strategies for boosting the action of PTEN in cancer.

The results obtained reveal PTEN activity to fluctuate, a phenomenon linked to its ubiquitination, a process regulated by the ubiquitin E3 ligase cCBL. It is shown that cCBL encounters PTEN after growth factor (GF) receptor activation and this is dependent on PI3-kinase. These observations highlight the cCBL-mediated cross-regulation between PI3-kinase and PTEN, and suggest a novel mechanism for the control of PTEN activity.

## 2. Materials and Methods

### 2.1. Cell Culture, Cell Activation, Reagents

HEK-293T (CRL-3216), PC3 (CRL-1435), and NIH 3T3 (CRL-1658) cells were purchased from the American Type Culture Collection and maintained DMEM (Gibco, Thermo Fisher Scientific, Waltham, MA, USA) plus 10% FCS, 2 mM glutamine, 10 mM HEPES, 100 IU/mL penicillin, and 100 μg/mL streptomycin. Murine embryonic fibroblasts (MEF) were obtained from wild-type C56BL/6 mice and maintained in DMEM supplemented with 10% FCS, 4 mM glutamine, 50 μM β-mercaptoethanol, 1 mM sodium-pyruvate, and antibiotics as above. All cells were grown at 37 °C in a 5% CO_2_ atmosphere. For short duration kinetics studies, all cells—except for MEF—were cultured without FCS (2 h) and then incubated with FCS (15%), EGF (100 ng/mL), or platelet-derived growth factor (PDGF, 50 ng/mL) for the indicated times. MEFs were grown to confluence and maintained confluent 24 h before serum starvation. The NAE1 inhibitor MLN-4924 was from MedChemExpress (Monmouth Junction, NJ, USA); PR-619, MG132, NEM, and the PTEN inhibitor BpV(PIC) were from Sigma-Aldrich (Saint Louis, MO, USA). MK2206 was from Tocris Bioscience (Bristol, UK).

### 2.2. cDNAs, siRNA, Transfection

pRK5-PTEN and pRK5-C124S-PTEN were kindly donated by R. Pulido (Ikerbasque, Bilbao, Spain) [13]. pRK5-PTEN-GST was prepared by eliminating the PTEN stop codon by site directed mutagenesis using the QuikChange II Kit (Agilent, Santa Clara, CA, USA). GST was PCR-amplified with primers including the Sal I and Apa I sites and then inserted at these sites at the PTEN C-terminus. The bacterial pGEX3x-PTEN vector (Addgene, Watertown, MA, USA) was used for in vitro ubiquitination assays. PTEN-Luc cDNA was prepared by amplifying PTEN from pEGFP-PTEN [14] using primers with the PTEN stop codon eliminated and EcoRI, BamHI flanking sites. The PCR product was inserted at these sites in pRLuc-N1. The constructs for energy transfer experiments were prepared as follows. PI3-kinase α (p110α) was amplified with primers including the XhoI and BamHI flanking sites. The products were inserted into pECFP-C1 and pEYFP-C1 at the corresponding sites. PI3-kinase β (p110β) was amplified with primers including the XhoI and EcoRI flanking sites, and the products similarly subcloned. p85α was amplified with the EcoRI and BamHI flanking sites; p85β and the p85α SH3 domain were amplified with EcoRI and the KpnI flanking sites, and all three subcloned into the appropriate sites in pECFP-C1 and pEYFP-C1. For the p85β SH3 domain, a BamHI restriction fragment was obtained from pEFBOS-XC-hHA-SH3 Bcr p85β [15] and subcloned into pECFP-C1 and pEYFP-C1. Other constructs used have been previously reported [16]. Short-hairpin RNA (shRNA) for UBC12 (in the pLKO- puromycin vector) was obtained from Sigma-Aldrich; control and PTEN-specific siRNA were from Invitrogen (Thermo Fisher). siRNA for mouse cCbl and human cCBL, CBLb, and NEDD4-1 (Smartpools) were from Dharmacon (Lafayette, CO, USA). cDNA transfection was performed using JetPei Polyplus, and siRNA transfection with Lipofectamine RNAiMax (Invitrogen). For energy transfer assays, cells were transfected with cDNA and polyethylenimine (5.47 mM) in 150 mM NaCl; this was added to cells in FCS-free medium for 4 h before transfer to complete medium.

### 2.3. Cell Harvesting, Cell Lysis, Western Blotting, Immunoprecipitation

Cells for harvesting were rinsed in PBS (137 mmol/L NaCl, 2.7 mmol/L KCl, 10 mmol/L Na_2_HPO_4_, 1.8 mmol/L KH_2_PO_4_) and scraped off in the presence of 1 mL of PBS. Cell suspensions were centrifuged (250 *g*, 2 min) and the pellets either immediately frozen or lysed. For pAKT-only analyses, tissue culture dishes were immediately frozen in dry ice until lysis. Different detergent buffers were used, all of which, however, contained protease and phosphatase inhibitors (1 mM PMSF, 1 mM Na_3_VO_4_, leupeptine 10 μg/mL, aprotinin 10 μg/mL, 5 mM NaF, 10 nM okadaic acid). To prepare whole cell extracts (WCE), the cells were lysed in RIPA buffer (Tris-HCl 20 mM pH 7.4, NaCl 137 mM, MgCl_2_ 1 mM, CaCl_2_ 1 mM, NP40 1%, Na-deoxycholate 0.5%, SDS 0.1%) (1 h, 4 °C). For PTEN sumoylation, NEM (1 μM) was added to RIPA buffer. For protein-interaction studies, cells were lysed in NP-40 buffer (10 mM Tris-HCl pH 7.4, 150 mM NaCl, 10 mM KCl, 1% NP-40) (immunoprecipitation [IP] of EGFR), or in Brij96 buffer (10 mM Tris-HCl pH 7.4; 150 mM NaCl; 1 mM EDTA; 1% Brij 96) (cCBL IP). All extracts were centrifuged (16,000 *g*, 15 min, 4 °C) and the protein concentration in the supernatant tested using the Bradford method or the MicroBCA Kit (Pierce, Appleton, WO, USA).

For IP, WCE were pre-cleared by incubation with protein A (Prot A) or Prot G-sepharose (4 °C, 1 h) (Thermo Fisher). Pre-cleared extracts were incubated with the appropriate antibody (see Ab list) at 4 °C (1–3 h) and incubated for 1 h more with Prot A or G (4 °C). For controls, Prot A/G beads were incubated with WCE without Ab, or with Ab but without WCE. Immunoprecipitates were washed three times with lysis buffer, three times with TBST buffer (TBS buffer [50 mM Tris-HCl, pH 7.5, 150 mM NaCl] containing 0.1% Tween 20), and three times with TBS. In the TBST washes, the IP were incubated for 10 min in an end-over-end rotor (4 °C). The IP were boiled in Laemli buffer, resolved by SDS-PAGE and examined by Western blotting (WB). For the latter, the resolved IP or extracts (50 μg) were transferred onto PVDF or nitrocellulose membranes and probed with the primary Ab and horseradish peroxidase-conjugated secondary Ab (Dako, Agilent). PVDF membranes were used for Ubiquitin and pAKT blots. Blots were developed with ECL reagent (GE Healthcare, Chicago, IL, USA). Immunoblotting band signals were quantitated using FIJI software (NIH, Bethesda, MD, USA).

### 2.4. Subcellular Fractionation

Harvested cells were immediately suspended in 250 μL of cytosol buffer (PBS, 1 mM MgCl_2_, phosphatase, and protease inhibitors, as above), frozen at −20 °C, and thawed at room temperature (RT°) (3 cycles). The final extracts were centrifuged (1 h, 20,000 *g*, 4 °C) and the supernatants saved as the cytosol-enriched fraction. The pellets were then suspended in 250 μL of plasma membrane buffer (50 mM Tris-HCl pH 8, 150 mM NaCl, 1 mM MgCl_2_, 0.5% NP-40, plus inhibitors, as above), incubated for 15 min at 4 °C, and centrifuged (15 min, 20,000 *g*, 4 °C). The final supernatants represent the plasma membrane-enriched fraction. The resulting pellets were suspended in 200 μL of RIPA buffer and incubated for 15 min at 4 °C before adding 20 μL of 10× DNAse A buffer (100 mM Tris-HCl pH7.5, 25 mM MgCl_2_, 5 mM CaCl_2_) and 100 units DNAse I (RT°, 30 min). These suspensions were sonicated, centrifuged (15 min, 20,000 *g*, 4 °C), and the final supernatants deemed to represent the nuclear protein-enriched fraction.

### 2.5. PTEN Sumoylation and Ubiquitination

To enrich extracts in ubiquitinated proteins, TUBES peptide columns were used as previously reported [17]. Briefly, ~4 × 10^6^ cells in p100 culture dishes were lysed in 500 μL of TUBES lysis buffer (10 mM Tris-HCl pH 7.4, 150 mM NaCl, 2.5 mM MgCl_2_, 0.5% NP-40, PR-619 50 μM) containing 100 μg of purified GST-TUBES fusion protein (or GST for the negative control) (15 min, 4 °C). The extracts were cleared by centrifugation (4 °C, 16,000 *g*, 15 min) and incubated with glutathione-Sepharose 4B beads (GE healthcare) (45 min, 4 °C). These beads were then collected by centrifugation (4 °C, 72 g, 5 min) and washed three times in TBST buffer and once in TBS. Bead-bound proteins were eluted by incubation (1 h, 4 °C) with 200 μL Tris-HCl 50 mM pH9 buffer containing 10 mM glutathione (Sigma-Aldrich). The eluted proteins were dialyzed in TUBES buffer (with only 0.1%NP-40) using cellulose membranes (Sigma-Aldrich D9277-100FT) (2 h, RT°).

For PTEN sumoylation and ubiquitination, RIPA cell extracts or dialyzed ubiquitin-enriched extracts, respectively, were incubated (90 min, 4 °C) with PTEN XP D4.3-Ab-beads or DA1E Isotype control beads (CST). They were then washed as above (see IP). PTEN ubiquitination or sumoylation in vivo was determined from the signal in ubiquitin or sumoylation blots, as measured using Fiji software (NIH).

### 2.6. PTEN Phosphatase Assay

For the PTEN activity assays, cells were lysed in a detergent buffer (10 mM Tris-HCl [pH 7.4], 150 mM NaCl, 10 mM KCl, 0.5% Nonidet P-40) with protease inhibitors (Roche Applied Science, Penzberg, Upper Bavaria, Germany), okadaic acid, and orthovanadate, but without NaF phosphatase inhibitor. Cells were incubated with lysis buffer at 4 °C for 1 h, then centrifuged (9300 *g*, 10 min). Cell extracts were pre-cleared by 2 h of incubation with Prot A beads (4 °C) and the resulting supernatant subjected to IP using 138G6 anti-PTEN Ab (Cell Signaling, Danvers, MA, USA) (4 °C, 3 h) and then Prot A (4 °C, 1 h). Duplicated IP were extensively washed. One set was examined by WB using the PTEN Ab, and the other set used for the phosphatase assay performed with the K-4700 Kit (Echelon Biosciences, Salt Lake City, UT, USA) following the manufacturer’s instructions. Briefly, lyophilized PI(3,4,5)P_3_ (diC16) substrate stock was dissolved (vortexed and sonicated) at low concentration (100 μM) in water. Purified PTEN was washed and suspended in 75 μL reaction buffer (50 mM Tris [pH8], 50 mM NaCl, 10 mM DTT, 10 mM MgCl_2_, 3 μM PI(3,4,5)P_3_). A PTEN-only control and a no-enzyme control were run in parallel. Reactions proceeded with shaking for 30 min at 37 °C. Since the PTEN was bound to beads, a short 100 g spin allowed for the separation of the enzyme and the product (the latter in the supernatant). The amount of PI(4,5)P_2_ produced in the reaction was measured in triplicate (25 μL/each) by ELISA (Echelon Biosciences), using a stranded curve with different amounts of PI(4,5)P_2_. The prepared ELISA assays were incubated (1 h at 37 °C) and the resulting color measured at 450 nm in a plate reader (Thermo Fisher). The values of the known PI(4,5)P_2_ standards were used to interpolate the values recorded in the PTEN test reactions. Phosphatase activity values were normalized to the amount of PTEN.

### 2.7. In Vitro Ubiquitination Assay

For cCBL phosphorylation, 50 ng of Y505F-Lck [18] were mixed with the cCBL IP (1 mg cell extract) or r-(recombinant) cCBL (200 ng) and incubated in Lck kinase buffer (25 mM Tris-HCl pH 7.4, 5 mM MnCl_2_, 50 μM ATP) for 15 min at 25 °C. cCBL reactions were performed in ubiquitination buffer (40 mM Tris-HCl pH 7.4, 2 mM DTT, 5 mM MgCl_2_, 5mM ATP). The reaction volumes (40 μL) contained 200 ng purified recombinant (r)-cCBL (Merck-Millipore, Burlington, MA, USA), 5 μg ubiquitin, 250 ng UBA1, and 250 ng of UBE2H5B or UBE2N/Ubc13 (all from Boston Biochem, Cambridge, MA, USA). Parallel reactions with cCBL and WWP1 (20 μL) were performed in ubiquitination buffer using r-cCBL (200 ng) or r-WWP1 (Ubiquigent, Dundee, Scotland, UK) (200 ng), UBE2H5B (300 ng), UBA1 (50 ng), and energy regeneration solution (Boston Biochem). To test for PTEN ubiquitination, GST-PTEN (100–400 ng, indicated) was purified by pull-down, suspended in ubiquitination buffer, and added to the reaction. Reactions (2 h, 30 °C) were terminated with 1× Laemli buffer.

### 2.8. Immunofluorescence, Videomicroscopy

IF studies were performed as previously described [19]. Briefly, cells were seeded on slides pretreated with 50 μg/mL Type I-collagen (Sigma-Aldrich). After activation with PDGF or FCS, the cells were fixed with 4% paraformaldehyde (RT°, 10 min) and then permeabilized in PBS with 0.3% Triton X-100 (RT°, 10 min), blocked with PBS plus 10% FCS and 0.01% TX100 (RT°, 90 min), and incubated with PTEN Ab (Millipore, A2B1), (1 h, RT°). Cy3 anti-mouse secondary Ab was used for detection (Jackson Laboratory, Bar Harbor, ME, USA). DNA was stained with Hoechst 33258 (Molecular Probes, Eugene, OR, USA). Images were acquired at 63× using an SP5 confocal microscope and employing Leica TCS SP5 software. PM-positive cells were selected by measuring the fluorescence signal (in arbitrary units) in a region of interest (ROI) at the PM. When this signal was at least 1.5 times that obtained in the same-size ROI for the adjacent cytosol, the cell was considered PM-positive (i.e., for PTEN). Cells with a similar signal in the PM and cytosol ROI were considered PM-negative. Cells were considered PM-intense when the integrated density of the PM ROI was ≥4 times that of the adjacent cytosol ROI.

For video microscopy, 15 × 10^3^ NIH3T3 cells were seeded onto µ-Slide 8 Well dishes (IBIDI). After 24 h, these cells were transfected with pEGFP-N1 or pEGFP-Btk-PH vectors (0.2 µg plus 0.4 µg JetPei/dish). On the next day, the cells were incubated in serum-free RPMI (without phenol red) for 2 h and examined using a Leica DMi8 S epifluorescence microscope equipped with an sCMOS Orca-Flash 4.0 camera and a live cell chamber maintained at 37 °C (5% CO_2_ atmosphere). An image was taken prior to activation (t = 0) and recording began after the addition of PDGF (50 ng/mL) or FCS (15%). The recording conditions were: Led 490 nm at 50%, exposure time 150 ms, objective 40×/0.8, magnification-changer 1.6×. An image was taken every 2 min for 90 min. Signal quantization was performed using FIJI software (NIH).

### 2.9. RT-PCR Analysis, Cell Cycle Progression, and Lentiviral Infection

RNA was extracted using the RNeasy Kit (Qiage, Germantown, MD). cDNA was synthesized using the high capacity cDNA reverse transcription kit (Applied Biosytems, Thermo Fisher). Quantitative real-time PCR was performed using Evagreen qPCR mix (Solis Biodyne, Tartu, Estonia). RQ values were normalized with GAPDH. Cell cycle progression was examined as described [20]. Briefly, the cells were incubated with 20 μM bromodeoxyuridine (BrdU, 1 h), washed and incubated in medium without BrdU, and at different times were stained with anti-BrbU-FITC Ab (BD Biosciences, Franklin Lakes, NJ, USA) and propidium iodide and examined by flow cytometry. For lentiviral production, HEK-293T cells were transfected using JetPei reagent with the pLKO-puromycin lentiviral plasmid in the presence of pMD2.G and psPax2 (Addgene) (48 h). Supernatants containing the viruses were filtered (0.45 μm pore size), supplemented with polybrene (8 μg/mL, Sigma-Aldrich), and added to the cells. Infection was repeated at 24 h. At 72 h, puromycin-resistant cells were selected with puromycin (2 μg/mL, 96 h) (Sigma-Aldrich).

### 2.10. Statistical Analysis

Statistical analyses were performed using GraphPad Prism (San Diego, CA, USA).

Fluorescence and bioluminescence resonance energy transfer, and the primer and antibody lists are included in Appendix A.

## 3. Results

### 3.1. EGF and Serum Trigger Reverse the Fluctuations in AKT and PTEN Activity

To study the change in endo-PTEN activity after the addition of GF, immortalized human HEK-293T cells, which express the WT forms of PTEN and PI3-kinase [21]), were examined. The cells were first preincubated without serum (2 h) to reduce the basal activation of intracellular signaling pathways, and then activated for short periods (0–120 min) using FCS. Endo-PTEN was immunoprecipitated from the activated cell extracts and its phosphatase activity determined by measuring the conversion of PIP_3_ into PIP_2_. PIP_3_ levels were monitored in parallel with Thr308pAKT and Ser473pAKT levels [1,22].

FCS triggered an early increase in Thr308pAKT (at ~5 min) followed by a reduction (at ~10–15 min), then a second peak (at ~30–60 min) and subsequent reduction (Figure 1a). In well-resolved gels, the Thr308pAKT signal was detected as two bands, with one band perhaps resulting from an additional Akt modification. Endo-PTEN activity was tested in parallel and a similar fluctuation seen, with a reduction at ~5 min and a second at ~30–60 min following the addition of FCS (i.e., coinciding with pAKT peaks) (Figure 1b). Indeed, the simultaneous representation of Thr308pAKT and PTEN activity revealed maximum Thr308pAKT levels to coincide with reductions in PTEN activity (Figure 1c). Optimal AKT activation is known to require the phosphorylation of AKT at Ser473 [23]. Ser473pAKT variations were similar but were detected slightly later than those of Thr308pAKT (Figure 1a, c). Over the 2 h examination period, two (sometimes three) complete fluctuation cycles (with inverse AKT and PTEN peaks and troughs) were detected.

Cells activated with EGF were examined similarly. As with FCS, EGF induced fluctuating AKT and PTEN activities (Appendix A). In contrast, the phosphorylation of ERK led to a single activation wave (Figure 1a, Appendix A). The phospho-EGFR signal also showed one major wave, although it returned at ~2 h due to recycled EGFR (Figure 1a). The PTEN, AKT, and PI3-kinase protein content remained constant over the interval examined, although at 120 min the PTEN level was moderately reduced (Figure 1b; Appendix A). The exact time of the pAKT peaks varied slightly in different assays (Appendix A) but in every one they coincided with the PTEN activity troughs (Figure 1c, Appendix A).

To determine whether these fluctuations also occurred in primary cells, a similar assay was performed using fresh MEF. As with the HEK-293T cells, FCS stimulation triggered Thr308pAKT signal fluctuations with two peaks at ~5 min and ~30–60 min (Figure 1d,e; Appendix A). Ser241PDK1 phosphorylation is known to be dependent on PIP_3_ [24]; the cells showed high Ser241PDK1 levels after serum starvation, but they also fluctuated after the addition of GF (Figure 1d,e). The Ser241PDK1 and PDK1 blots showed two bands perhaps reflecting more than one modification of PDK1. Ser473pAKT levels also fluctuated in serum-activated MEF (Figure 1d,e), while PTEN, AKT, and PI3-kinase protein levels remained unchanged (Figure 1d). These results show that Thr308pAKT levels and PTEN activity fluctuate in GF-activated primary cells, and that PIP_3_/pAKT peaks coincide with PTEN activity troughs.

### 3.2. PDGF and Serum Induce Transient PIP_3_ Recruitment to the Plasma Membrane

In an attempt to confirm the PIP_3_ fluctuations at the PM following growth factor receptor (GFR) activation, NIH3T3 cells expressing green fluorescent protein (GFP) fused to the Btk–pleckstrin homology (PH)-domain were subjected to live imaging. Since the Btk-PH domain binds selectively to PIP_3_ [25], it is possible to examine PIP_3_ levels at the cell membrane [26]. Nuclear PIP_3_ levels were not assessable since GFP-Btk-PH (like GFP) constitutively localizes to the nucleus [26].

NIH3T3 cells (immortal embryonic fibroblasts) are highly adherent and flat, and therefore optimal for video microscopy. They showed pAKT fluctuations upon addition of FCS or PDGF (see below). In control cells transfected with GFP, no apparent changes in GFP localization were detectable after adding FCS or PDGF (Appendix A). In contrast, in cells expressing GFP-Btk-PH, both FCS and PDGF induced a rapid recruitment of the PH domain (that binds to PIP_3_) at the PM (Appendix A). This first Btk-PH burst (within minutes) at the membrane was sharper after the addition of PDGF than after FCS. PDGF binds to a single receptor type resulting in synchronous PI3K activation, while FCS activates different GF receptors (those of LPA, insulin, sphingosine 1-P, etc.) with different activation kinetics. In the videos, the second PIP_3_ peak was subtler, possibly due to the basal PIP_3_ levels detected on cell extensions throughout the study period. Nonetheless, a higher, sustained Btk-PH recruitment at the cell border was detectable in several sequential frames during the 30–60 min observation period, corresponding to the second pAKT peak (Figure 2a,b; Appendix A). Different cells showed slightly different times of appearance for the first and second PIP_3_ peaks; in most cases the first peak was detected in the very first minutes (1 to 5), and the second around 30–70 min (Figure 2; Appendix A, representative of *n* = 20 videos with each GF). Quantitation of the Btk-PH signal in a virtual line across the PM (Figure 2) confirmed greater recruitment of Btk-PH at two discrete time points, confirming the presence of the first and second PIP_3_ bursts.

### 3.3. pAKT Fluctuations Require PTEN Expression

Given the complementary patterns of AKT and PTEN activity, it was postulated that PTEN inactivation might help maximum PIP_3_ levels be reached (and maximal AKT activity) upon the addition of GF. This hypothesis was tested in normal MEF by the depletion of PTEN with siRNA, which abrogated the fluctuation in pAKT in response to FCS (Figure 3a).

In a reverse approach, after confirming that cancer cells lacking PTEN expression (PC3 prostate cancer cells) showed a single AKT activation wave after the addition of serum (Figure 3b), the consequences of reconstituting PTEN expression were investigated. PTEN expression was examined using different recombinant (r)-PTEN cDNAs, and PTEN-Luc levels were found to be within the range of endo-PTEN (Appendix A). PTEN-Luc expression in *PTEN−/−*PC3 cells restored the pAKT fluctuation and increased the size of the pAKT peaks compared to those recorded for control PC3 cells (Figure 3b, top). The overexpression of PTEN to 10× normal using PRK5-WT-PTEN also restored pAKT fluctuations, and reduced the mean pAKT levels (Figure 3b, bottom). These results show that PTEN activity induces fluctuations in pAKT.

The requirement of PTEN phosphatase activity for pAKT fluctuations was confirmed by comparing PC3 cell reconstitution with WT or inactive-C124S-PTEN. Reconstitution with C124S-PTEN yielded a single-wave pAKT pattern, as seen in *PTEN* null PC3 cells (Appendix A). Additionally, the treatment of HEK-293T with a PTEN inhibitor (BpV) prior to FCS stimulation flattened pAKT levels (Appendix A).

### 3.4. pAKT/PTEN Fluctuations Accelerate Cell Division

To determine whether abrogating the AKT fluctuations had any functional consequence, the cell division time and the progression of the cell cycle of control *PTEN−/−*PC3 cells and WT-PTEN-reconstituted cells were compared. The PTEN-reconstituted PC3 cells (~13 h) had a shorter duplication time than the *PTEN−/−* cells (~18 h, *n* = 3). Calculation of the time spent in each phase (from the percentage of cells in each phase and t1/2), revealed the *PTEN−/−* cells to enter the S phase earlier, but they were delayed in G2/M (Appendix A).

Cells were incubated with BrdU (1 h) in the S phase and then deprived of BrdU to examine the progress of the cell cycle. Analysis of the BrdU signal vs. the DNA content confirmed that *PTEN−/−* cells entered the S phase earlier, but were delayed at G2/M to G0/G1 transition (Figure 3c). These alterations were also clear in plots of the number of cells remaining in G0/G1 (at 1 h) or G2/M (at 7–9h) after BrdU deprivation (Figure 3d, Appendix A). PTEN might affect the progression of the cell cycle in a phosphatase-independent manner [27], but since constitutive PI3-kinase expression induces a similar phenotype as PTEN loss [20], the present results support the idea that PTEN-mediated PI3-kinase/AKT fluctuations are required for optimal cell cycle progression.

### 3.5. Maximum pAKT Levels Correlate with PTEN Ubiquitination

To study the mechanism of PTEN inactivation, a number of post-translational modifications (PTMs) were taken into account. PTEN SUMO-1 modification increases PTEN localization at the PM [11], whereas C-terminal PTEN phosphorylation (pPTEN) and PTEN ubiquitination can inactivate PTEN [10,28].

No substantial changes in endo-PTEN CT-phosphorylation were detected in HEK-293T cells stimulated with FCS for 0–90 min (Figure 4a). To examine ubiquitination, endo-PTEN was immunoprecipitated from FCS-activated HEK-293T extracts and examined in blots. This revealed a fluctuation in endo-PTEN ubiquitination after FCS stimulation although the ubiquitination signal was very weak (Appendix A). To improve the detection of ubiquitinated-PTEN, the ubiquitinated proteins in whole cell extracts (WCE) were concentrated in Tandem Ubiquitin-Binding-Entities (TUBES) peptide columns [17] (diagram in Appendix A). PTEN was immunoprecipitated from ubiquitin-enriched extracts and examined in blots. The addition of FCS increased ubiquitinated-PTEN levels at ~5 min, coinciding with the first pAKT peak. They then fell before increasing again at the time of the second pAKT peak and remained high at later time points (60–90 min) (Figure 4b). Subsequent PTEN blots yielded a nearly indistinguishable pattern (Figure 4b). The ladder pattern obtained in these blots suggests that, in some of the PTEN Lys residues, multiple ubiquitins had been incorporated. 

PTEN sumoylation was examined in blots of PTEN immunopurified from WCE. Sumoylation was also transient, and did not coincide with PTEN ubiquitination or the pAKT peaks, but co-occurred with a reduction in pAKT (Figure 4b). Subsequent PTEN blots identified sumoylated and non-modified PTEN (Figure 4b). PTEN secondary blots also identified a fraction of PTEN to be sumoylated prior to stimulation; this was better detected in SUMO2,3 blots (see below). Therefore, after the addition of FCS, PTEN underwent early ubiquitination coinciding with the maximum pAKT levels (and low PTEN activity, Figure 1). Posterior sumoylation coincided with the fall in pAKT levels.

### 3.6. FCS, PDGF, and EGF Addition Alters PTEN Localization at pAKT Peaks and Troughs

Since PTEN ubiquitination and sumoylation might alter PTEN localization [11,12], tests were made to see whether PTEN modifications induced by FCS concurred with changes in PTEN localization. Cells were activated with FCS or EGF for short time periods (0–90 min) and then subjected to subcellular fractionation and the PTEN in the cell fractions examined by WB. At all times, the majority of the PTEN signal was found to remain in the cytosol (>80%) (Figure 5a). In contrast, the amount of PTEN at the PM was found to be reduced at 5 and 30-to-60 min after the addition of FCS or EGF, when pAKT levels were at their highest. PTEN relocated back to the PM later, when pAKT levels were low (Figure 5a, Appendix A). Moderate increases in nuclear-PTEN content were found at the times when PTEN detached from the PM; in the nuclei a higher MW band of potentially modified PTEN was detected (Figure 5a, Appendix A).

The changes in PTEN localization were confirmed by immunofluorescence (IF). PTEN Ab specificity (in IF) was tested by PTEN depletion in human and mouse cells; only in the latter cells did PTEN depletion reduce the PTEN IF signal (Appendix A). The IF analysis was therefore performed in murine NIH3T3 cells. These were cultured for 2 h without serum and then activated with FCS or PDGF, which induced pAKT fluctuation with maximum levels at 5 and 30 min (Figure 5b). The proportion of cells with a PTEN signal at the PM was reduced at 5 and 30 min, i.e., when the pAKT signal was strong (Figure 5b). At these times, a moderate increase in the nuclear PTEN signal was detected. In contrast, at 10 and 60-to-90 min (when pAKT levels were low), PTEN localized to the PM in a large proportion of the cells (Figure 5b). Therefore, cell activation with FCS, PDGF, or EGF reduced PTEN levels at the PM at ~5 min, coinciding with the times when pAKT levels were highest (and PTEN-ubiquitinated); PTEN relocation to the PM occurred in tandem with low pAKT levels (and PTEN sumoylation).

### 3.7. cCBL Is Essential for FCS-Induced Early PTEN Ubiquitination

WWP1, WWP2, and NEDD4.1 (all HECT E3 ligases that ubiquitinate PTEN) [29,30,31,32] were deemed potential candidates for modulating PTEN ubiquitination after GF addition. However, while WWP2 is known to induce PTEN degradation [29], this was not seen in the present work shortly after the addition of GF (Figure 1). Further, WWP1 expression is driven by MYC, which is not induced early after GF addition [30]. Thus, only NEDD4.1 was examined. Since PTEN inactivation coincides with AKT activation, E3 ligases regulated by PI3-kinase/AKT, such as CBL and cullin ubiquitin E3 ligases [15,33], were also examined. Cullin E3 ligases require neddylation by NAE1 (NEDD8 activating E1) and UBC12 (ubiquitin conjugating enzyme 12) to be active [34]. The consequences of depleting UBC12 or of inhibiting NAE1 (with MLN-4924 [35]) were therefore examined. UBC12 depletion was checked by qPCR, and NAE1 inhibition by examining p27kip1 levels, which are regulated by cullin E3 ligases [33]. Interference with cullin E3 ligase activity by NAE1 inhibition or UBC12 depletion did not alter PTEN ubiquitination after the addition of GF (Figure 6a).

The involvement of NEDD4-1, cCBL, and CBL-b in PTEN ubiquitination was examined using siRNA. Whereas NEDD4-1 depletion did not alter PTEN ubiquitination or the changes in pAKT after FCS addition, both cCBL and CBL-b depletion reduced PTEN ubiquitination soon after FCS addition and flattened the Thr308pAKT levels (Figure 6b). Since PTEN sumoylation takes place after PTEN ubiquitination (see above), the effect of cCBL depletion on sumoylation was also examined. cCBL depletion reduced the small PTEN sumoylation levels seen in quiescence (clearly detected with SUMO2,3 Ab) and abrogated PTEN sumoylation after GF addition (Appendix A). These observations show that cCBL is essential for the early ubiquitination of PTEN after cell stimulation with GF, and for later PTEN sumoylation.

### 3.8. Purified cCBL Ubiquitinates Purified PTEN In Vitro

Since cCBL was more abundant than CBL-b in the HEK-293T cells, the mechanism of PTEN regulation by CBL molecules was studied by focusing on cCBL. Tests were made to determine whether purified cCBL ubiquitinated purified PTEN in vitro. A ubiquitination reaction was first optimized using cCBL purified from HEK-293T cells; cCBL was tested prior to and upon its Tyr phosphorylation by an active form of the Src-family kinase Lck (Y505F-Lck) [18] (Appendix A). Phospho (p)-cCBL or non-modified cCBL were incubated with ubiquitin, E1 ligase, and E2 ligases (UBE2H5B or UBE2N) in an ATP-containing buffer. This reaction revealed the cCBL auto-ubiquitination capacity, which was greater for p-cCBL (Appendix A).

To test for PTEN ubiquitination by cCBL, r-cCBL (from baculovirus), and PTEN purified from bacteria were used (Appendix A). r-cCBL ubiquitinated PTEN in vitro (Figure 6c). The high MW band detected in PTEN blots even in the absence of cCBL (Figure 6c, d) might correspond to bacterially aggregated PTEN. The capacity of cCBL to modify PTEN was compared to that of WWP1; cCBL ubiquitinated PTEN in vitro to an extent similar to WWP1 (Figure 6d). PTEN ubiquitination can reduce PTEN phosphatase activity [28]; indeed, cCBL-mediated ubiquitination in vitro induced an ~20% reduction in PTEN phosphatase assays (Appendix A). Together, cCBL expression is essential for in vivo PTEN ubiquitination at ~5 and ~30–60 min after the addition of GF. It is also required for later PTEN sumoylation. The cCBL ubiquitination of PTEN might be direct, as purified r-cCBL was able to ubiquitinate r-PTEN in vitro.

### 3.9. PI3-Kinase Controls cCBL Association with EGFR

To test whether the addition of GF brings PTEN and cCBL into proximity, cells were activated with EGF and the recruitment of cCBL or PTEN to EGFR examined by IP and immunoblotting. As PTEN band resolves close to the IgG heavy chain (used in IP), analyses were performed using PTEN-Luc (which has a higher MW). The low PTEN levels increase by PTEN-Luc expression in HEK-293T cells did not affect pAKT fluctuation (Appendix A). EGFR IP followed by PTEN blotting showed that a fraction of PTEN was bound constitutively to EGFR (Figure 7a, left); in contrast, EGF induced a transient translocation of cCBL to EGFR (middle). A small fraction of cCBL was constitutively bound to PTEN (Figure 7a, right); this basal complex looks not be bound to EGFR since no cCBL was found associated with EGFR prior to stimulation (Figure 7a, middle). Thus, EGFR binds to PTEN constitutively and to cCBL only after EGF addition. 

EGF is known to induce PI3-kinase translocation to EGFR [36]. Since the PI3-kinase regulatory subunit (p85^PI3K^) binds to cCBL after T cell activation [15], it was postulated that EGF might trigger a p85^PI3K^-mediated recruitment of cCBL to EGFR. Biochemical examination of the endogenous p85^PI3K^/cCBL complex was difficult due to the non-specific bands of p85^PI3K^ Ab in blots. The involvement of p85^PI3K^ in cCBL recruitment to EGFR was tested by modification of the p85α and p85β levels (Appendix A). p85^PI3K^ overexpression (indicated as ↑p85α p85β) enhanced, and p85^PI3K^ depletion (↓p85α p85β) abrogated, cCBL association to EGFR (Figure 7b). The EGFR/PTEN and cCBL/PTEN complex levels, however, were not proportional to p85^PI3K^ levels (Appendix A).

Since a fraction of PTEN is bound constitutively to EGFR, EGF should bring p85^PI3K^ into proximity with PTEN. To test this, IP/WB was avoided given the non-specific p85α Ab bands in blots. Instead, PTEN/p85^PI3K^ proximity was examined by fluorescence energy transfer (FRET) [37,38,39]. The efficiency of FRET in detecting p85^PI3K^ complexes was confirmed using fluorescent forms of the PI3-kinase catalytic and regulatory subunits (using the SH3-p85^PI3K^ domain as a negative control) (Appendix A). FRET also confirmed the formation of a PI3-kinase α and β complex [21] mediated by p85α ^PI3K^ but not by p50α^PI3K^ (Appendix A).

Energy transfer between p85α ^PI3K^ and PTEN was examined taking advantage of the PTEN-Luc bioluminescence signal. Titration of the bioluminescence energy transfer (BRET) between PTEN-Luc and YFP-p85α ^PI3K^ revealed a positive interaction (Figure 7c). Moreover, the addition of EGF induced a transient increase in the BRET signal at 5–15 min showing that EGF increases p85α ^PI3K^ and PTEN proximity (Figure 7d).

It was postulated that the depletion of cCBL by reducing PTEN ubiquitination would impair PTEN detachment from the PM. The cCbl-depleted NIH3T3 cells showed a modest but significant increase in the proportion with PM-bound PTEN compared to controls (Figure 7e, Appendix A).

Thus, EGF stimulates a p85^PI3K^-dependent translocation of cCBL to EGFR, where a fraction of PTEN is constitutively bound. Once close, cCBL can modulate PTEN ubiquitination/inactivation and its detachment from the PM, thus allowing maximum PIP_3_/pAKT levels to be reached. After PTEN detachment from the PM, PTEN undergoes a sumoylation process that likely induces its later membrane localization and the reduction of pAKT levels simultaneous with PTEN sumoylation. The capacity of PI3-kinase to bring cCBL to the receptor and thus induce transient PTEN inactivation and internalization reveals that crosstalk occurs between the enzymes regulating PIP_3_ (PI3-kinase and PTEN) and highlights cCBL as a new target for the control of PTEN activity.

## 4. Discussion

The aim of this study was to understand how PTEN activity is regulated after cell activation by growth factors, and to determine the mechanism linking PI3-kinase and PTEN activities for controlling PIP_3_ levels. It was found that both the activity of the PI3-kinase effector AKT and the membrane PIP_3_ levels fluctuate following the addition of GF, with a first peak at ~5 min and a second one at ~30–60 min. Between these peaks, membrane PIP_3_ levels and pAKT levels decrease. The time frame under analysis was chosen by taking into account previous studies showing that PTEN is active at the membrane at ~90 min following the addition of GF [21]. Within this period, PTEN showed activity fluctuations complementary to those of AKT. When AKT was activated, PTEN was inhibited, and vice versa, suggesting that AKT and PTEN activity variations are probably linked. Indeed, PTEN expression was essential for the fluctuation in AKT activity. PTEN underwent a ubiquitination event early after the addition of GF, at the time of maximum pAKT levels. cCBL expression was required for PTEN early ubiquitination in cells activated by GF. So, either PTEN is ubiquitinated directly by cCBL (as cCBL is able to ubiquitinate PTEN in vitro), or cCBL brings a ubiquitin E3 ligase to PTEN. The ubiquitination of PTEN reduced its activity and correlated with its detachment from the membrane. These results show that maximum PIP_3_/pAKT levels require not only PI3-kinase to be active but also the inactivation of PTEN. It is here shown that the addition of EGF triggers the PI3-kinase-dependent recruitment of cCBL to the EGFR, where it encounters PTEN. Thus, PI3-kinase induces PTEN inactivation by bringing the cCBL E3 ligase to the EGFR, which in turn results in PTEN ubiquitination.

Although the transient nature of AKT activation by GF was known, the fluctuation and the complementary behavior of PTEN and AKT activity, were not. Most previous studies included a single or two-three activation time points [2,21,22]. Dalle Pezze et al. [40] did, however, include a detailed kinetic study (as we do here) using HeLa cells activated with insulin; their blots also showed Thr308pAKT fluctuation, but the authors passed over this observation as their focus was mTOR.

AKT/PTEN fluctuation was detected in normal murine and immortalized fibroblasts, in human PTEN-reconstituted PC3 cells, and in HEK293T cells, showing that it is a generalized process. It could be argued that the two pAKT peaks (in the 0–90 min timeframe) could be linked to two different inputs of PI3-kinase activation, such as by GFR and GTPase activation [16,41]. However, these inputs do not explain the reductions in pAKT levels. The present assessment of PTEN phosphatase activity showed that PTEN activity also fluctuates, and that pAKT modulation requires PTEN activity. PTEN depletion abrogated AKT fluctuations, which were rescued by expression of WT-PTEN (but not inactive-PTEN) in PTEN−/− cells. These transient fluctuations in pAKT are not irrelevant since the depletion of PTEN (here) and the expression of a constitutive, active PI3-kinase [20] both eliminate them, resulting in delayed exit from G2/M and slower progression through the cell cycle.

One of the best-established PTEN regulatory mechanisms is the phosphorylation of its C-terminal cluster of four Ser and Thr residues, which keeps PTEN inactive and stable [10,13]. In the present work, phospho-PTEN levels remained constant after the addition of FCS (0–90 min), ruling out CT-phosphorylation as the mechanism behind PTEN inactivation after the addition of GF. The present work does not, however, exclude a potential reduction in the phosphorylation of individual CT-Ser and Thr residues since the Ab used for the analysis does not distinguish fully phosphorylated PTEN from single residue phosphorylation defects [42].

Ubiquitination might also reduce PTEN activity [28]. Indeed, PTEN ubiquitination at ~5 min after GF addition concurred with PTEN showing low phosphatase activity. PTEN ubiquitination was unaffected by interference with cullin E3 ligases or by NEDD4 depletion, but it was nearly abrogated by cCBL or CBL-b depletion. cCBL and CBL-b also cooperate for optimal GFR downregulation [43]. The involvement of NEDD4 in PTEN ubiquitination in vivo has been controversial [31,32] and the presented observations support that NEDD4 is not involved in GF-induced early PTEN ubiquitination. Since purified cCBL was able to ubiquitinate purified PTEN in vitro, cCBL might ubiquitinate PTEN directly. Future descriptive studies will examine which PTEN Lys residues are modified in a cCBL-dependent manner.

The reduction in pAKT levels correlated with an increase in PTEN sumoylation and PTEN localization to the PM. PTEN binding to PIP_2_ in the PM triggers an allosteric activation of the phosphatase [44] which could account for the increase in PTEN activity once localized to the PM. PTEN sumoylation seems to also require the former PTEN ubiquitination process since cCBL depletion reduced PTEN ubiquitination and sumoylation.

To gain insight into how cCBL and PTEN might encounter one another, the association of these molecules, and with EGFR, was examined. Whereas a fraction of PTEN was constitutively bound to EGFR, cCBL recruitment to EGFR only occurred after the addition of EGF. Cell activation induces PI3-kinase translocation to GFR [36] but also cCBL binding to p85^PI3K^ [15]. Here it was postulated, and confirmed, that cCBL recruitment to EGFR (where PTEN is bound) is controlled by p85^PI3K^. This model was challenged by testing two predictions made in view of these results. The first was that the addition of EGF should also bring p85^PI3K^ and PTEN into proximity. This was confirmed using bioluminescence energy transfer assays. The second was that, since cCBL depletion abrogates early GF-induced PTEN ubiquitination, which correlated with its detachment from PM, cCBL depletion might increase PTEN localization at the PM. This was confirmed by immunofluorescence.

It remains to be determined whether cCBL and p85^PI3K^ stay bound to PTEN in its transit to the nucleus. Previous studies have shown that p85^PI3K^, p110α, p110β, and PTEN all form part of a large (~600-kDa) macromolecular complex (identified by gel filtration), supporting the idea that PTEN and p85^PI3K^ might transit to the nucleus in complex [21,45]. cCBL might also internalize bound to EGFR and PTEN.

## 5. Conclusions

The present results reveal a mechanism for the control of endogenous PTEN activity upon the addition of GF. Shortly after its addition, the activity of the PI3-kinase effector AKT, and that of PTEN, show complementary fluctuations in behavior. PTEN is needed for these fluctuations since its depletion results in the flattening of pAKT levels. Moreover, PTEN reconstitution in PTEN-deficient cells restores pAKT fluctuation, which is needed for optimal progression through the cell cycle. AKT/PTEN co-regulation involves the EGF-induced recruitment of PI3-kinase to EGFR, and in turn that of cCBL. The expression of cCBL E3 ligase is essential for PTEN ubiquitination in cells. cCBL mediated PTEN ubiquitination in vitro reduced its phosphatase activity and correlated with PTEN detachment from the PM. The present results show that cCBL recruitment to the EGFR is mediated by PI3-kinase. Thus, by inducing cCBL translocation to the EGFR, PI3-kinase promotes the inactivation of PTEN allowing maximum PIP3/pAKT levels to be reached. After cCBL-regulated PTEN ubiquitination, PTEN undergoes a process of sumoylation and re-localizes to the membrane, a process that occurs with a simultaneous reduction in pAKT levels. The biological significance of these results is that the crosstalk of PI3-kinase and PTEN, mediated by cCBL, guarantees both the optimal and transient nature of PIP_3_ bursts. Based on these results, the reduction of PTEN ubiquitination, and the boosting of PTEN sumoylation, might help to increase PTEN activity in cancer cells.

## Figures and Tables

**Figure 1 cells-10-02803-f001:**
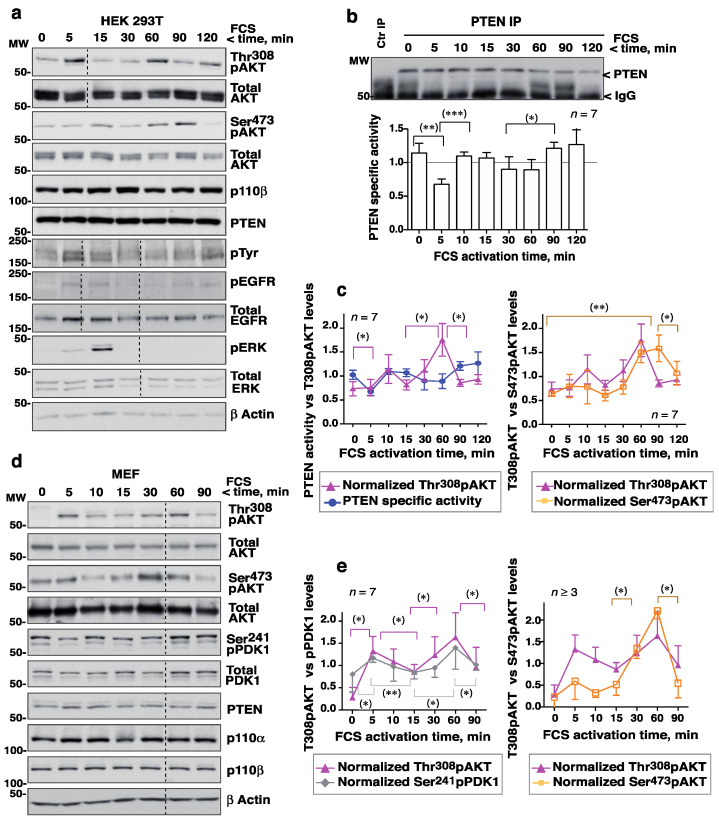
Activation of cells with FCS triggers complementary fluctuations in pAKT levels and PTEN activity. (**a**–**c**) HEK-293T cells were incubated in serum-free medium (2 h) then activated with FCS (15%) over different lengths of time (indicated). (**a**) Cells were lysed and normalized extracts examined by WB with different Ab (indicated). (**b**) Cell extracts as in (**a**) (1 mg) were incubated with PTEN Ab for immunoprecipitation (IP). Purified PTEN was examined in a phosphatase assay that measures the PTEN phosphatase product PIP_2_. The blot confirmed efficient PTEN IP. The graphs show PTEN activity as the PIP_2_ produced by PTEN at each time point (in arbitrary units), normalized for PTEN content. To compare phosphatase activity at different times, the PIP_2_ levels at each time were normalized to the mean PIP_2_ level (equivalent to 1). For controls (Ctr IP), extracts were incubated with protein A. (**c**) The pAKT signal from assays as in (**a**) was measured, corrected for the AKT and β−actin content, and normalized to the mean pAKT signal for each assay (equivalent to 1). The graphs represent the relative PTEN activity and pAKT levels (Thr308 or Ser473) at each time point (mean ± SD, *n* = 7). (**d**) MEF were grown to confluence, maintained confluent for 24 h, and then incubated in FCS-deprived medium (2 h) before treatment with FCS (15%) for different times. Extracts were examined as in (**a**). (**e**) The graph shows the Ser473pAKT, Thr308pAKT and Ser241pPDK1 signals normalized to AKT and PDK1 and to β−actin (mean ± SD, *n* = 7). Dashed lines indicate time points left out from the original gels (originals blots are included as Appendix A). * *p* < 0.05, ** *p* < 0.01, *** *p* < 0.001, Student’s paired *t* test.

**Figure 2 cells-10-02803-f002:**
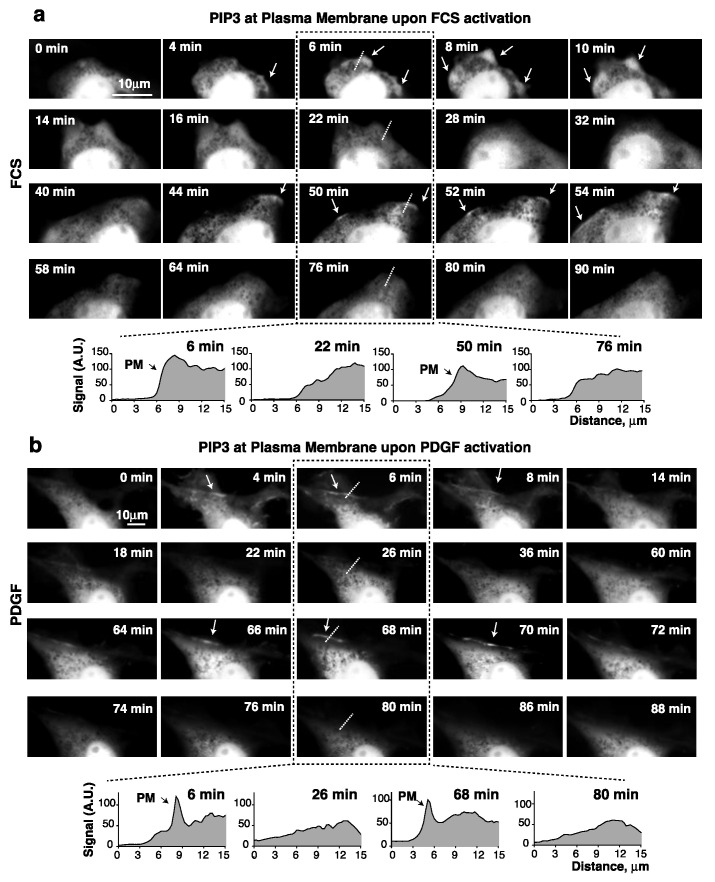
PDGF and serum induce PIP_3_ recruitment at the plasma membrane. (**a**,**b**) NIH3T3 cells were transfected with GFP-Btk-PH (48 h), then incubated in serum-free medium (2 h) and activated with FCS (15%) (**a**) or PDGF (50 ng/mL) (**b**). An image was taken prior to activation (t = 0) and then every 2 min for 90 min after GF addition. Representative frames from the video (time after GF addition indicated). The graphs at the bottom (**a**,**b**) show the Btk-PH signal (in arbitrary units) on the virtual line indicated in the central images.

**Figure 3 cells-10-02803-f003:**
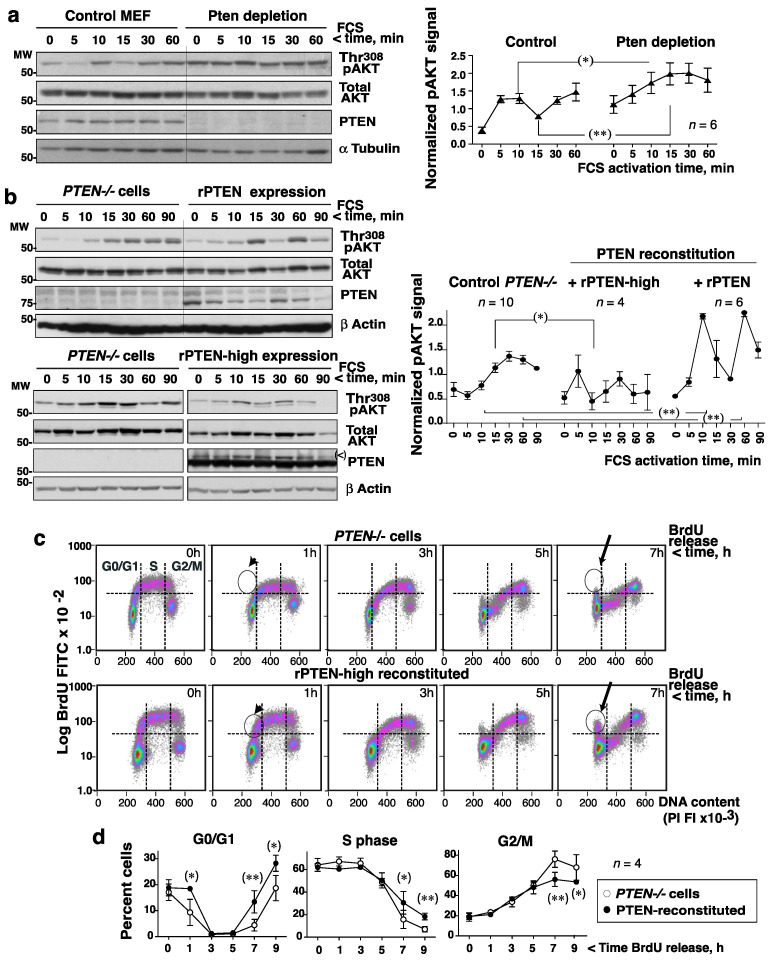
pAKT fluctuations require PTEN expression and affect cell cycle progression. (**a**) MEF transfected with a control or a *Pten* siRNA (72 h) were grown to confluence, incubated without FCS (2 h), then treated with FCS (15%) for different times. Blots show the PTEN and Thr308pAKT levels at different time points. The graphs represent the relative pAKT signal (corrected for AKT and α−tubulin) normalized to the mean pAKT signal for control MEF (considered as 1) (mean ± SD, *n* = 6). (**b**) PC3 cells were transfected with control, PTEN-Luc, or pRK5-PTEN (high expression, indicated) cDNAs (48 h) and then activated with FCS as in (**a**). Blots and graph plotting were as in (**a**). The upper band in the PTEN blot (indicated (<)) comes from the previous Akt blot. (**c**,**d**) Control or pRK5-PTEN-reconstituted PC3 cells (as in b) were labeled with BrdU (20 μM, 1 h), then deprived of BrdU, and incubated for different times. (**c**) DNA content of the BrdU+ cells as determined by flow cytometry. Arrowheads point at BrdU+ cells in the G1-S phase; arrows point to the BrdU+ cells arriving at G0/G1 after G2/M (**c**). (**d**) The graphs show the proportion of cells in each phase at different times (mean ± SD, *n* = 4). * *p* < 0.05, ** *p* < 0.01 Student’s unpaired *t* test.

**Figure 4 cells-10-02803-f004:**
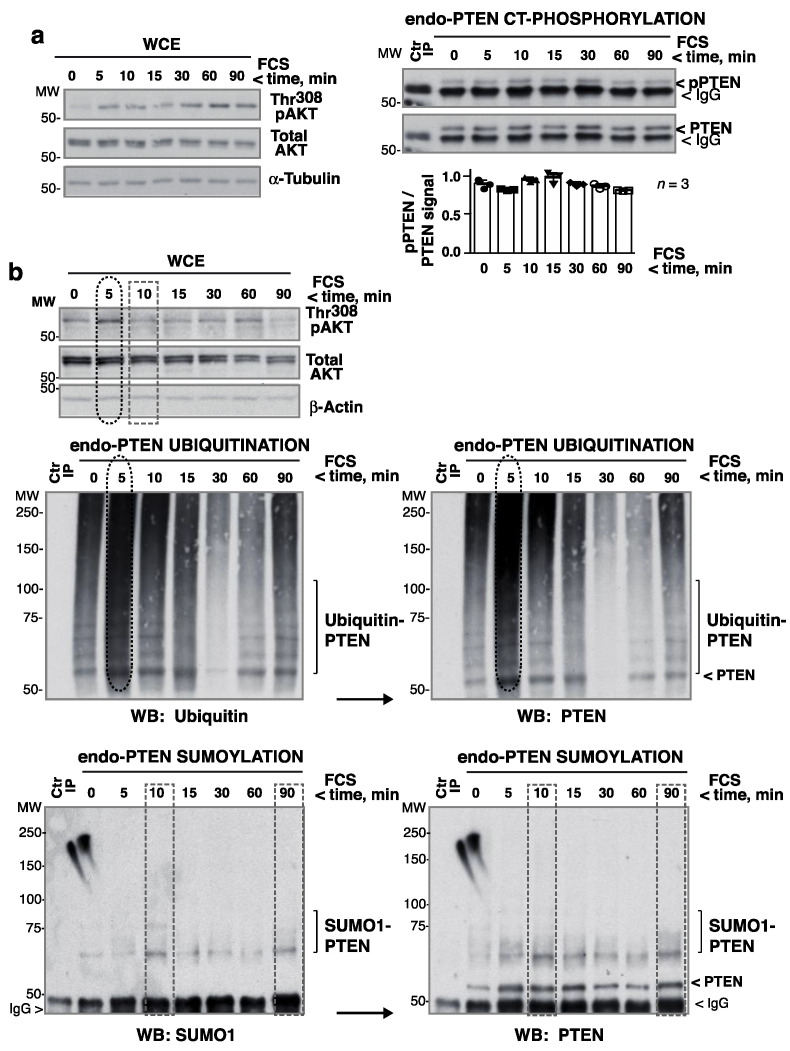
Maximum pAKT levels correlate with PTEN ubiquitination. (**a**) HEK-293T cells were incubated in medium without serum (2 h), and then stimulated with FCS (15%) for different times. Left blots show Thr308pAKT levels in WCE. Extracts (1 mg) were also incubated with anti-PTEN Ab for IP and pPTEN levels determined in WB (right panels). For controls (Ctr), extracts were incubated with Prot A. The graph shows the normalized pPTEN signal (corrected for PTEN levels) with respect to the mean pPTEN signal (considered 1) (mean ± SD, *n* = 3). (**b**) HEK-293T cells were activated as in (**a**); Thr308pAKT levels were examined in WCE (top panels). Extracts were also enriched for ubiquitinated proteins in TUBES columns. The eluted extracts were incubated with PTEN Ab-beads for IP and these examined by WB (top left). Secondary PTEN blots confirmed the retention of more ubiquitinated-PTEN at 5 min in the TUBES columns (right). PTEN was also immunoprecipitated from WCE and its sumoylation examined by WB. Subsequent PTEN blots showed the efficiency of PTEN IP. For controls, extracts were incubated with control beads. Maximum ubiquitination (ellipses) and sumoylation (rectangles) are indicated.

**Figure 5 cells-10-02803-f005:**
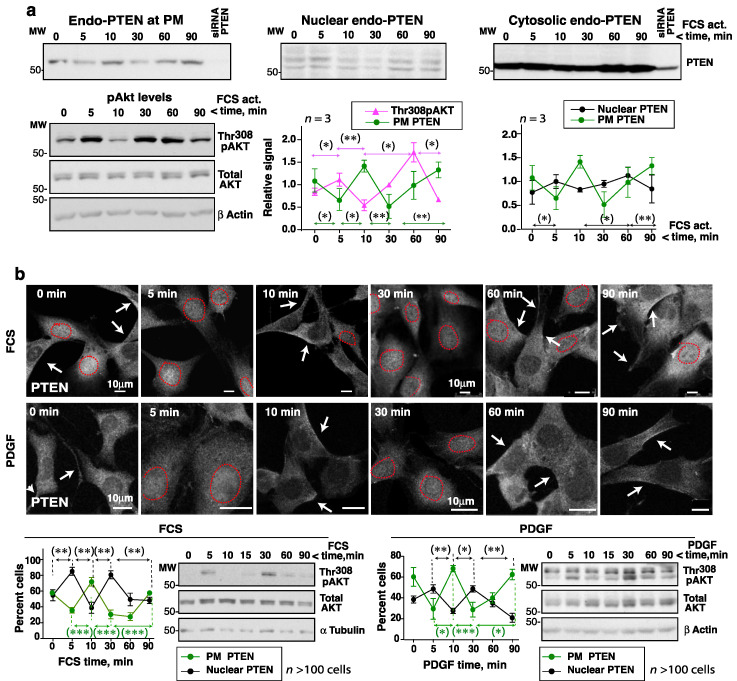
Cell activation with FCS or PDGF alters PTEN localization at the times of pAKT peaks and troughs. (**a**) HEK-293T cells were incubated in serum-free medium (2 h) and then stimulated with FCS (15%). The pAKT content in WCE was tested in a fraction of the cells (indicated). The remaining cells were subjected to fractionation to isolate the cytosol, PM, and nuclear fractions. Blots show the PTEN levels in the different fractions. The graphs show the pAKT and PTEN signals corrected for the AKT and the cell fraction marker contents, respectively, and normalized to the mean pAKT or PTEN signal (equivalent to 1) (mean ± SD, *n* = 3). (**b**) Representative IF images from an assay (of *n* = 4 with similar results) showing the PTEN signal in NIH3T3 cells incubated in serum-free medium (2 h) and then stimulated with FCS (15%) or PDGF (50 ng/mL) for different times. A fraction of the cells was lysed and the pAKT levels at different times examined by WB. The graphs show the percentage of cells with a PM PTEN-positive signal (arrows), or a PTEN-nuclear positive signal (surrounded by a dashed red line) (mean ± SD, *n* > 100). (**a**,**b**) * *p* < 0.05, ** *p* < 0.01, *** *p* < 0.001 Student’s *t* test.

**Figure 6 cells-10-02803-f006:**
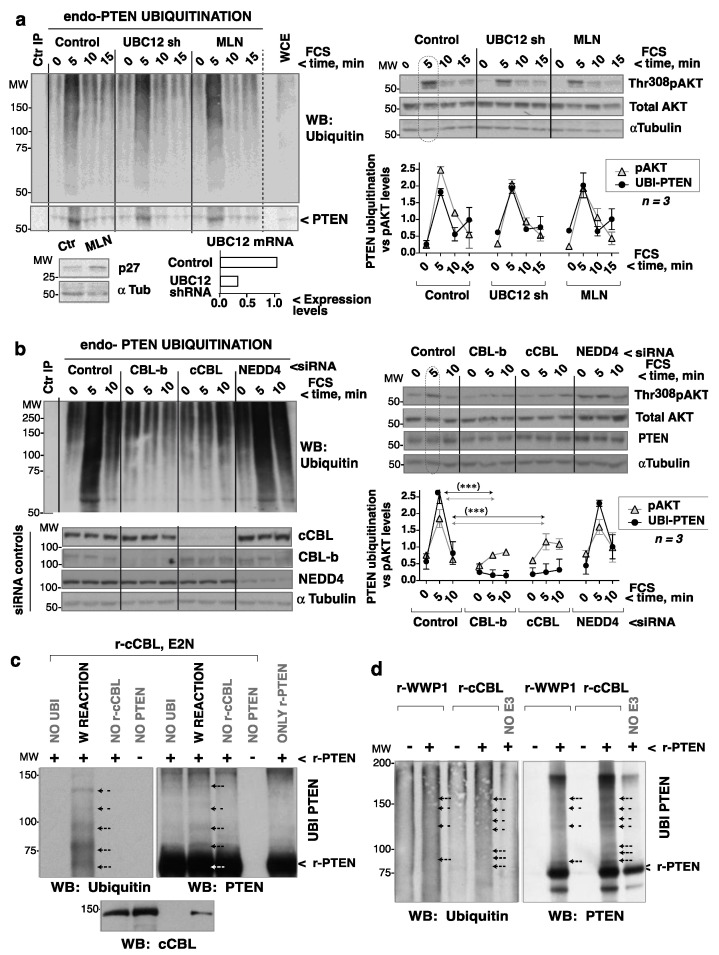
cCBL depletion abrogates GF-induced PTEN ubiquitination. (**a**) HEK-293T were infected with control- or UBC12 shRNA-encoding lentiviruses (72 h), or treated with MLN-4924 (1 μM, 48 h), then incubated in serum-free medium (2 h) before stimulating with FCS (15%) (indicated). Thr308pAKT levels were examined in WCE. Cell extracts were enriched for ubiquitinated proteins by purification in TUBES columns. The eluted ubiquitin-enriched extracts were used for PTEN IP with Ab-fused Sepharose beads. For controls (Ctr IP), extracts were incubated with isotype-matched control Ab beads; PTEN ubiquitination was examined by WB. Dashed lines indicate lanes that were cut out from the original blots (included as Appendix A). UBC12 shRNA efficiency was assessed by q-PCR; MLN-4924 activity via the examination of p27kip levels upon cell incubation without FCS (16 h). pAKT signals were measured, corrected for the AKT level, and normalized to the mean pAKT signal in controls (equivalent to 1). The endo-PTEN ubiquitination signal (60–120 KDa) was measured, corrected for cellular PTEN levels (in WCE), and compared to the mean PTEN ubiquitination signal (considered 1). The graph shows the pAKT signal vs. the ubiquitinated PTEN signals (mean ± SD, *n* = 3). (**b**) HEK-293T cells were transfected with control, *NEDD4-1, cCBL,* or *CBL-b* specific siRNA pools (72 h). Cell processing, blots, and graphs are as in (**a**). (**b**) *** *p* < 0.001 Chi squared test; the PTEN and pAKT significance levels were similar. (**c**) Duplicate reactions containing purified r-cCBL (200 ng) and GST-PTEN (200 ng) were incubated with ubiquitin, E1 ligase (250 ng), and UBE2N ligase (250 ng) in ubiquitination buffer (whole reaction; W REACTION). Some of the reactions were performed in the absence of ubiquitin, PTEN, or cCBL (indicated). (**d**) Duplicate reactions containing non-phosphorylated r-cCBL or r-WWP1 (200 ng) were incubated in the presence or absence of r-PTEN (100 ng), as well as ubiquitin, E1 (50 ng), UBE2H5B (300 ng), and energy regeneration solution. (**c**,**d**) Duplicate reactions were examined in parallel WB.

**Figure 7 cells-10-02803-f007:**
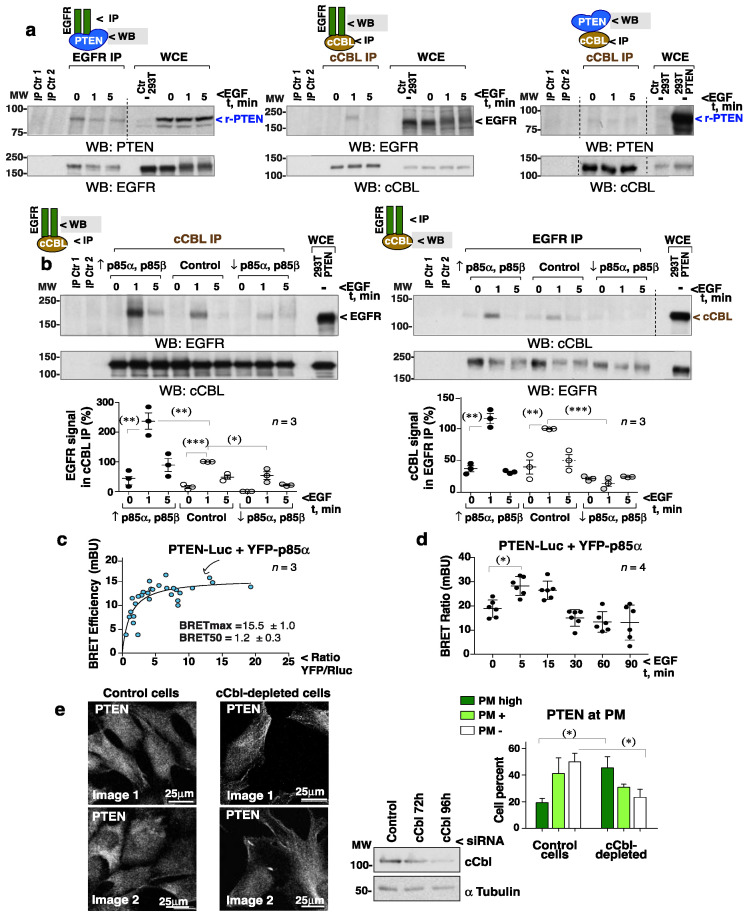
EGFR associates with cCBL in a PI3-kinase-dependent manner. (**a**,**b**) HEK-293T cells were transfected with PTEN-Luc alone (48 h) (**a**), or with a mixture of PTEN-Luc, HA-p85α, and HA-p85β cDNA (48 h) (indicated as ↑ p85α, p85β), or were first transfected with a mix of *PIK3R1* (p85α) plus *PIK3R2* (p85β) siRNA (72 h) and then with PTEN-Luc (last 48 h) (↓ p85α, p85β) (**b**). The efficiency of these transfections was tested by WB (Appendix A). Cells were then incubated in serum-free medium (2 h) and stimulated with EGF (100 ng/mL). NP-40 or Brij96 extracts (1 mg) were used for EGFR and cCBL IP, respectively. For controls, Prot A or G were incubated with Ab (IP Ctr 1) or lysate (IP Ctr 2). The presence of associated proteins in EGFR or cCBL immunoprecipitates were examined by WB. The last lanes show control WCE (indicated). The graphs (**b**) show the EGFR signal in cCBL IP, corrected for the cCBL content and normalized to the maximum EGFR signal in complex with cCBL (control at 1 min, 100%) (mean ± SD, *n* = 3). The graph for the cCBL signal in EGFR IP was similarly plotted (*n* = 3). (**a**,**b**) Dashed lines indicate the cutting out of lanes from the original blots (included as Appendix A). * *p* < 0.05, ** *p* < 0.01, *** *p* < 0.001, Student’s unpaired *t* test. (**c**) BRET saturation curves in HEK-293T cells transfected with a constant amount of PTEN-Luc and increasing amounts of YFP-p85α. Data were fitted to a non-linear regression equation assuming one binding site (*n* = 3). FRETmax/BRETmax and FRET50/BRET50 values (mean ± SD) were calculated using a non-linear regression equation (*n* = 3). (**d**) BRET efficiency in HEK-293T cells transfected with a fixed 1:1 amount of PTEN-Luc and YFP-p85α (48 h) and stimulated with EGF (100 ng/mL) for the indicated times (mean ± SD, *n* = 4) (*) one-way ANOVA. (**e**) Exponentially growing control and cCbl-depleted NIH3T3 cells (72 h) were transferred to glass coverslips (24 h) prior to the IF detection of PTEN (two representative images, each). Blots confirmed cCbl depletion efficiency. The graph illustrates the percentage of cells (mean ± SD, *n* = 100 cells) with an intense, positive, or negative PTEN PM signal (indicated). * *p* < 0.05, ** *p* < 0.01 Student’s unpaired *t* test.

## Data Availability

All original data are included as a Appendix A.

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
