# Peer review of "Fluctuations in AKT and PTEN Activity Are Linked by the E3 Ubiquitin Ligase cCBL"

_cells, 2021, doi:10.3390/cells10112803_

Round 1
Reviewer 1 Report
The manuscript by Olazabal-Moran et al. provides a detailed characterization for the kinetics of AKT and PTEN activity control upon stimulation, namely EGF and serum addition. In particular, they report a mutual exclusive activation for each component that fluctuates during the response. Critically, through a systematic analysis, they identified as the functional link for these 2 opposing activities the cCBL E3-ubiquitin ligase that is required for PTEN ubiquitination, elimination from the plasma membrane and inactivation. Subsequently, SUMOylation of PTEN (that also depends on cCBL) allows the re-association of PTEN to plasma membrane. The PI3-kinase was also shown as a required component for the association of cCBL with PTEN.
This is a rather comprehensive/detailed analysis that provides new insights into the control of the AKT/PTEN module. While the kinetics for the AKT/PTEN activation have been addressed in previous studies, the functional link for this phenomenon remained unknown. Thus, the identification of cCBL as the potential E3-ligase that co-ordinates AKT/PTEN activities is an important advance in the field. The authors have used multiple complementary approaches and the experiments are well presented and convincing.
Specific comments:
-Regarding the cell cycle analysis the authors as model system the PTEN knockout cells with PTEN re-expression. While a useful approach, the issue is that these cells may have adaptive responses. In addition, the analysis is performed under unstressed conditions (No EGF, serum addition). If the authors want to link the cell cycle effects with the observed oscillation in AKT/PTEN activity, they may consider the use of PTEN inhibitors and acute PTEN inhibition upon stimulation.
-The ePTEN term they use to indicate endogenous PTEN may be confusing as previous studies have reported the “engineered” PTEN (e-PTEN) with enhanced activites.
-The manuscript will benefit from careful proof-reading.
Author Response
Specific comments:
-Regarding the cell cycle analysis the authors use as model system the PTEN knockout cells with PTEN re-expression. While a useful approach, the issue is that these cells may have adaptive responses. In addition, the analysis is performed under unstressed conditions (No EGF, serum addition).
If the authors want to link the cell cycle effects with the observed oscillation in AKT/PTEN activity, they may consider the use of PTEN inhibitors and acute PTEN inhibition upon stimulation.
Answer: The advantage of the cell cycle analysis under unstressed conditions (in cells growing normally in serum containing media) is that it provides a cell cycle distribution closer to normal conditions. For this reason, even being more difficult and time-consuming, the BrdU pulse chase in PTEN-deficient or PTEN-reconstituted cultures was considered the optimal test to examine the differences in cell cycle progression.
In addition, the conclusion on the need of pAKT fluctuations for optimal cell cycle progression was based not only on the analysis performed here (comparing PTEN-deficient versus PTEN-reconstituted cells) but also on the coincidence of the cell cycle defects with those detected in cells expressing a constitutively active form of PI3K, which also eliminates pAKT fluctuations (see page 10, 3rd paragraph).
Nonetheless, intrigued by the well-taken point that acute PTEN inhibition might or might not have similar effects than its depletion or deletion, we tested the consequences of treating the cells with PTEN inhibitors just prior to GF addition; PTEN inhibition also reduced pAKT fluctuations (new Supplementary Figure 3b). This analysis confirmed that PTEN inhibition also impaired pAKT fluctuations, as PTEN depletion or deletion.
-The ePTEN term they use to indicate endogenous PTEN may be confusing as previous studies have reported the “engineered” PTEN (e-PTEN) with enhanced activities.
Answer: The point is well taken; we have changed e-PTEN by endo-PTEN throughout the paper.
-The manuscript will benefit from careful proofreading.
Answer: A native-English speaker has revised the manuscript again. He has considered that in the title “fluctuation” is better than “oscillation”.
Reviewer 2 Report
Carrera and co-workers investigate the role of cCBL in the regulation of AKT and PTEN activities. Thorough experiments have been done to support the conclusion. I am supportive of the publication of this manuscript with a few clarifications listed below.
- In Figure 1a, 1d., there are several signals where two bands are detected, for example, 1d, Ser241 Ppdk1, please clarify.
- Is PTEN mono-ubiquitylated? Any insights into ubiquitylation sites?
Author Response
Carrera and co-workers investigate the role of cCBL in the regulation of AKT and PTEN activities. Thorough experiments have been done to support the conclusion. I am supportive of the publication of this manuscript with a few clarifications listed below.
- In Figure 1a, 1d. there are several signals where two bands are detected, for example, 1d, Ser241 Ppdk1, please clarify.
Answer: In the case of pAKT we indicated:
“In well-resolved gels, Thr308pAKT signal was detected as two bands; the extra band might result from an additional Akt modification” (page 6, 2º paragraph).
For PDK1 we have included:
“Ser241PDK1 and PDK1 immunoblots also detected two bands, which might reflect the existence of further modifications apart from Ser241 phosphorylation” (page 6, last paragraph).
- Is PTEN mono-ubiquitinated? Any insights into ubiquitination sites?
Answer: With regards to PTEN mono-ubiquitination, the ladder pattern obtained when PTEN immunoprecipitates were resolved and examined in blots using anti-ubiquitin antibodies suggest that at least in some of the Lys residues on PTEN, there is an incorporation of multiple ubiquitins (i.e. Fig. 4).
We have added:
““The ladder pattern obtained in these blots suggests that at least in some of the PTEN´s Lys residues, there is an incorporation of multiple ubiquitins” (page 10, last paragraph).
With regards to the second point, we have performed several proteomic analyses with the two mass spectrometry (ME) teams mentioned in acknowledgements. We encountered a number of difficulties. First, a low proportion of the cellular PTEN is ubiquitinated (most of the PTEN is cytosolic and most likely is not ubiquitinated). Thus, ubiquitinated peptides were of low abundance. Part of the problem might be that the ubiquitinated residues might suffer a fast de-ubiquitination in the cells. Other authors have included deubiquitinases (DUBs) inhibitors in the culture media; however, the inclusion of DUB inhibitors in the culture media was found to render PTEN ubiquitination stable throughout the kinetic impairing pAKT oscillations. Thus, in agreement with the model, the inclusion of DUBS inhibitors affected AKT and PTEN fluctuations, most likely because these oscillations require ubiquitination-deubiquitination cycles.
The worse limitation of the ME analysis was that PTEN has a very high number of Lys and Arg residues, which are susceptible of trypsin proteolysis (for ME analysis) and trypsin treatment produced many small peptides (i.e. the one encompassing K13), which were non-detected in ME. Although alternative proteases were tested, these yielded poorer results. On the opposite side, other peptides (i.e. the one containing K298) were enormous and also undetectable by ME in our hands. Some authors have either included exogenous mutations to be able to shorten too long peptides (i.e. K298-containing peptide; DOI: 10.1016/j.cell.2006.11.040). Other authors have mutated one by one all the Lys residues in PTEN to define which Lys residue mutant is no-longer modified by a given E3 ubiquitin ligase (DOI: 10.1038/s41467-020-15578-1). The latter approach will be taken in consideration to identify the PTEN's Lys whose ubiquitination is regulated by cCBL.
Round 2
Reviewer 1 Report
The revised version has addressed the main raised concerns. Additionally I support the idea of replacing oscillations with fluctuations as more accurate description.